# Genome-Wide Identification of Vitellogenin Gene Family and Comparative Analysis of Their Involvement in Ovarian Maturation in *Exopalaemon carinicauda*

**DOI:** 10.3390/ijms25021089

**Published:** 2024-01-16

**Authors:** Jiajia Wang, Shuai Tang, Qianqian Ge, Qiong Wang, Yuying He, Xianyun Ren, Jian Li, Jitao Li

**Affiliations:** 1Key Laboratory for Sustainable Utilization of Marine Fisheries Resources, Ministry of Agriculture and Rural, Yellow Sea Fisheries Research Institute, Chinese Academy of Fishery Sciences, Qingdao 266071, China; wangjj@ysfri.ac.cn (J.W.); 18837625451@163.com (S.T.); wangqiong@ysfri.ac.cn (Q.W.); heyy@ysfri.ac.cn (Y.H.); renxy@ysfri.ac.cn (X.R.); bigbird@ysfri.ac.cn (J.L.); 2Function Laboratory for Marine Fisheries Science and Food Production Processes, Qingdao National Laboratory for Marine Science and Technology, Qingdao 266237, China; ge2682@126.com

**Keywords:** vitellogenin, ovarian development, vitellogenesis, hepatopancreas

## Abstract

Vitellogenin (*Vtg*) is a precursor of yolk proteins in egg-laying vertebrates and invertebrates and plays an important role in vitellogenesis and embryonic development. However, the *Vtg* family remains poorly characterized in *Exopalaemon carinicauda*, a major commercial mariculture species found along the coasts of the Yellow and Bohai Seas. In this study, 10 *Vtg* genes from the genomes of *E. carinicauda* were identified and characterized. Phylogenetic analyses showed that the *Vtg* genes in crustaceans could be classified into four groups: Astacidea, Brachyra, Penaeidae, and Palaemonidae. *EcVtg* genes were unevenly distributed on the chromosomes of *E. carinicauda*, and a molecular evolutionary analysis showed that the *EcVtg* genes were primarily constrained by purifying selection during evolution. All putative *EcVtg* proteins were characterized by the presence of three conserved functional domains: a lipoprotein N-terminal domain (LPD_N), a domain of unknown function (DUF1943), and a von Willebrand factor type D domain (vWD). All *EcVtg* genes exhibited higher expression in the female hepatopancreas than in other tissues, and *EcVtg* gene expression during ovarian development suggested that the hepatopancreas is the main synthesis site in *E. carinicauda*. *EcVtg1a*, *EcVtg2*, and *EcVtg3* play major roles in exogenous vitellogenesis, and *EcVtg3* also plays a major role in endogenous vitellogenesis. Bilateral ablation of the eyestalk significantly upregulates *EcVtg* mRNA expression in the female hepatopancreas, indicating that the X-organ/sinus gland complex plays an important role in ovarian development, mostly by inducing *Vtg* synthesis. These results could improve our understanding of the function of multiple *Vtg* genes in crustaceans and aid future studies on the function of *EcVtg* genes during ovarian development in *E. carinicauda*.

## 1. Introduction

Vitellogenin (*Vtg*) is a major precursor of vitellin, which is a source of yolk nutrients for the development of ovaries and embryos in almost all oviparous organisms [1]. *Vtg* is generally present as a female-specific protein that binds to sugar, phosphorus, and lipids and acts as the major nutrient and energy source for early embryogenesis. In addition, an increasing number of studies have reported several non-nutritional roles of multiple *Vtg* genes, including in antioxidant stress responses and antibacterial activity [2,3,4]. The *Vtg* protein belongs to the lipid transporter superfamily and shares a conserved structural domain including a vitellogenin N-terminal domain, also known as the lipoprotein N-terminal domain (LPD_N), a domain of unknown function (DUF1943), and a von Willebrand factor type D domain (vWD) [5]. LPD_N is required for interaction with the *Vtg* receptor in *Macrobrachium rosenbergii* [6], which promotes the transport of Vg to oocytes, and also plays a conserved role in receptor recognition in other vertebrates and invertebrates [7]. DUF1943 and vWD play critical roles in pathogen recognition [8,9,10]. These studies indicate that *Vtg*, in addition to its involvement in yolk protein formation, also possibly plays non-nutritional roles for its conserved structural domain.

The process of yolk synthesis and accumulation in oocytes, known as vitellogenesis, plays a crucial role in the reproductive success of oviparous species. The *Vtg* gene family comprises various paralogs, exhibiting species–specific differences in both the structure and quantity of *Vtg* proteins among different oviparous vertebrates [1]. Most fish possess a tripartite *Vtg* system (VtgAa, VtgAb, and VtgC) in which all three forms of *Vtg* make a different contribution to the yolk. Different types of *Vtg* are essential and have disparate requisite functions at different times of development in fish [11]. The VtgAa is largely degraded into free amino acids during ovarian maturation. In contrast, the major yolks derived from VtgAb and VtgC remain largely intact and serve as yolk nutrients for later-stage embryos and larvae [12]. Genome editing reveals reproductive and developmental dependencies on specific types of vitellogenin in zebrafish (Danio rerio) [13]. Understanding the differences between variation of *Vtg* in mechanisms of yolk formation and processing is a benefit for the regulation of ovarian maturation and embryos and larvae development. Similar to in fish, previous studies on *Metapenaeus ensis* [14] and *Pandalopsis japonica* [15] have suggested the presence of multiple *Vtg* genes in these crustaceans, while most studies on the function of the *Vtg* gene in crustaceans have concentrated on a single *Vtg* gene. Meanwhile, the function of different vitellinogen genes in nutritional and non-nutritional roles in crustaceans has been little studied. Thus, considering their function in reproduction, the identification of the *Vtg* gene family in crustaceans and the analysis of their functions in ovarian development hold significant importance.

The ridgetail white shrimp, *Exopalaemon carinicauda*, belonging to the Palaemonidae family of crustaceans, is a major commercial mariculture species that is naturally distributed along the coasts of the Yellow Sea and Bohai Sea [16]. In addition to its economic value, *E. carinicauda* is a useful experimental organism in crustacean developmental biology because of its transparent body, large eggs, high reproductive capacity, and short reproductive cycle of only two to three months [17,18]. Female shrimps undergo at least three consecutive ovarian developmental stages during the reproductive cycle, and the ovaries of some shrimps gradually atrophy into cystic tubes after the first ovarian maturation stage. Several practical problems related to reproductive biology need to be solved in *E. carinicauda*. The precocious puberty usually occurred in the aquaculture of *E. carinicauda*, which affects its normal growth rate. The nonsynchronous ovarian development and spawning increase the consumption of manpower and financial resources of seeding. It is vital to better understand the regulatory mechanisms of vitellogenesis and ovarian development in *E. carinicauda*. The *Vtg* gene plays an important role in ovarian and embryo development in crustaceans [19], thus the *Vtgs* must be identified and clarified in *E. carinicauda*.

In recent years, many crustacean genomes have been sequenced and made available, providing a foundation for the identification and characterization of gene families at the whole-genome level [20,21]. However, only the *Vtg*-family genes of *M. rosenbergii* have been identified and characterized [22]. Selective regulation of multiple-Vtg uptake and processing during oocyte growth and ovarian maturation is critical to egg quality and developmental success. Differences in gene function and regulation amongst the different *Vtg* might have important physiological or biological effects [23]. There is significant variability in functioning of *Vtg* family members among the same tissue of different species, and in different tissues and developmental stages of the same species [12,24]. In *E. carinicauda*, only a single *Vtg* transcript has so far been identified [19]. The *Vtg* structure remains unknown, and as does whether multiple Vtgs are present in the *E. carinicauda* genome with contribution to vitellogenesis or not. Therefore, genome-wide identification, and investigation of the *EcVtg*-family genes in the *E. carinicauda* genome were performed in this study. The combination of long-read Pacific Biosciences and Hi-C technologies enabled the successful assembly of a high-quality genome sequence in *E. carinicauda* in our previous study. The final assembly covered 5.86 Gb, which was submitted to the NCBI genome database (PRJNA1055619). Firstly, the evolutionary relationships, gene structures, conserved domains, chromosomal locations, and motifs were analyzed to comprehensively understand the differences in the structure and potential function of *Vtg* family members using the genome sequences. In addition, we investigated gene expression patterns in different tissues and different embryo-development stages to identify their nutritional or non-nutritional roles. Finally, we performed the expression profile during ovarian development and expression changes after bilateral eyestalk ablation to evaluate their contribution to reproduction in *E. carinicauda*. This is the first systematic identification and functional investigation of *EcVtgs* in *E. carinicauda*. The results of this study will provide a basis for an improved understanding of the evolutionary and biological functions of *Vtg* genes in crustaceans, particularly in the context of ovarian development.

## 2. Results

### 2.1. Identification and Physicochemical Characterization of the EcVtg Gene Family

Ten *Vtg* genes were identified in the *E. carinicauda* genome using a combination of BLASTp and hidden Markov model (HMM) analyses and were confirmed using NCBI-CDD and SMART. The *Vtg* genes of *E. carinicauda* were named *EcVtg1–8*, according to the nomenclature of *M. rosenbergii* Vtgs. The results of BLASTX showed that *EcVtg1–7* were the most homologous to *Vtg1-7* in *M. rosenbergii*, whilst *EcVtg8* showed approximately 47.07–54.51% similarity with other *EcVtgs* (Appendix A).

The basic physicochemical properties of the proteins encoded with the *EcVtg* genes in *E. carinicauda* using he ExPASy tool (http://www.expasy.ch/tools/pi_tool.html, Swiss Institute of Bioinformatics, Lausanne, Switzerland) are shown in Table 1 and Appendix A. Bioinformatic analysis showed that the 10 *EcVtg* proteins contained 2421 (*EcVtg5*) to 2581 (*EcVtg8*) amino acids; the predicted molecular weights (MWs) ranged from 275.06 kDa (*EcVtg5*) to 291.37 kDa (*EcVtg2*); and the theoretical isoelectric point (pI) ranged from 7.27 (*EcVtg6b*) to 9.14 (*EcVtg1a*). The grand-average hydropathicity values of *EcVtg* proteins were all negative, indicating that all proteins were hydrophobic. The predicted protein-instability index (II) showed that all *EcVtg* proteins were unstable. In addition, the signal peptide prediction results demonstrated that all *EcVtgs* encoded proteins with Sec/SPI signal peptides had a likelihood of 0.9773–0.9983, and these gene products were preliminarily identified as secretory proteins.

The prediction of subcellular localization (using the online Wolf PSORT tool, https://www.genscript.com/wolf-psort.html, GenScript Biotech Corporation, Nanjing, China) suggested that most *EcVtg* proteins in *E. carinicauda* resided in the extracellular space. However, proteins encoded with *EcVtg1b*, *EcVtg4*, and *EcVtg7* were found to localize in the endoplasmic reticulum, while EcVtg8 was observed in the nucleus (Table 2). Additionally, the secondary structures of the *EcVtg* proteins were predominantly composed of alpha helices (38.00–40.09%), followed by random coils (35.35–37.26%). Extended strands and beta turns collectively accounted for 19.54–20.25% and 4.44–5.25% of the structural composition, respectively (Table 2).

### 2.2. Chromosome Location and Phylogenetics of the EcVtg Gene Family

To better understand the mechanism underlying the genomic distribution of *EcVtgs* on *E. carinicauda* chromosomes, a chromosome map of *EcVtgs* was constructed based on the genomic sequences of *E. carinicauda* (Figure 1). The results showed that all *EcVtg* genes were anchored on five chromosomes of the *E. carinicauda* genome, and the number of *EcVtgs* on the chromosomes ranged from one to seven. Chromosome 10 contained the largest number of genes (five), followed by chromosome 23 with two *EcVtg* genes.

To clarify the phylogenetic relationships and classification of *Vtg* genes among different species, a phylogenetic tree was constructed using *Vtg* protein sequences from *E. carinicauda* and other representative species (Figure 2 and Appendix A). According to the results, 94 *Vtg* genes were classified into five distinct groups: insecta, mollusks, fish, birds, and crustaceans, in accordance with the classic taxonomic structure. The crustacean group was subdivided into four distinct clusters: Astacidea, Brachyra, Penaeidae, and Palaemonidae. All EcVtg proteins converged with the Palaemonidae clade. The names of the members of the *E. carinicauda EcVtg* gene family were confirmed according to the branches of the phylogenetic tree and multiple sequence alignment analysis. The number of the *EcVtg*-gene-family members in *E. carinicauda* was similar to that in *M. rosenbergii* and *Macrobrachium nipponensis*. The *EcVtg* family of *E. carinicauda* contained eight members; that of *M. rosenbergii* contained seven members, except *Vtg8*; and that of *M. nipponensis* contained six members, except *Vtg5* and *Vtg6*.

### 2.3. Gene Structures and Conserved Motifs of the EcVtg Gene Family

The investigation into the *EcVtg* conserved motifs and exon–intron structures aimed to deepen our understanding of the conservation and diversification within the *EcVtg* genes using TBtools. As depicted in Figure 3, the 10 EcVtg proteins were classified into two subfamilies, aligning with the outcomes of the unrooted phylogenetic tree (Figure 3a). Genes with closer relationships exhibited significant structural conservation; however, variations were noted among different EcVtg proteins. For instance, *EcVtg1a*, *EcVtg1b*, *EcVtg3*, *EcVtg6a*, *EcVtg6b*, and *EcVtg8* contained 30 motifs, while *EcVtg4* and *EcVtg7* contained 9 motifs, excluding motifs 20 and 5, respectively. Moreover, *EcVtg5* displayed the fewest number of motifs. Specifically, *EcVtg2* contained 28 motifs, excluding 22 and 29.

The count of exons within the *EcVtg* family ranged from 15 to 17 (Figure 3b). *EcVtg4* possessed the highest number of exons, whereas *EcVtg3* and *EcVtg7* contained 16 exons. In contrast, the remaining *EcVtg* members consisted of 15 exons each.

### 2.4. Gene Duplication and Synteny Analysis

The Ka/Ks ratios were computed to explore the evolutionary constraints and potential selective pressures on the identified duplicated gene pairs (Appendix A) using KaKs_Calculator 2.0. The findings indicated that the Ka/Ks ratio values for the tandemly and segmentally duplicated *EcVtg* gene pairs ranged from 0.1981 to 0.4303, all of which were less than 1.0. This suggests that repetitive *EcVtg* genes in *E. carinicauda* were predominantly subjected to purifying selection during their evolutionary course.

### 2.5. Expression of EcVtg Genes in Different Tissues and Developmental Stages

To explore the possible functions of *EcVtgs*, their expression profiles in different tissues and different developmental stages were characterized using qRT-PCR and RNA-seq data, respectively. As shown in Figure 4, the expression patterns of *EcVtgs* in different tissues were different and tissue specific. All *EcVtg* genes showed preferential expression in hepatopancreatic tissues, especially *EcVtg1a*, *EcVtg1b*, *EcVtg2*, *EcVtg3*, *EcVtg5*, *EcVtg6b*, and *EcVtg7*, which exhibited much higher expression than that in other tissues (*p* < 0.05), suggesting that these gene members tend to play roles in vitellogenesis. Meanwhile, six members (*EcVtg2*, *EcVtg3*, *EcVtg4*, *EcVtg6a*, *EcVtg6b*, and *EcVtg7*) were expressed at significantly higher levels in the ovaries than in other tissues (*p* < 0.05).

As shown in Figure 5, except for *EcVtg4*, *EcVtg5*, *EcVtg6a*, and *EcVtg6b*, six EcVtgs were expressed at various levels based on the transcriptome databases of six different embryonic development stages and the first stage of zoea. *EcVtg1a*, *EcVtg1b*, and *EcVtg2* were expressed at high levels during the metazoea stage, which is the final stage of embryonic development in *E. carinicauda*. *EcVtg7* and *EcVtg8* were expressed most highly in the nauplius stage, and the highest expression of *EcVtg3* was in the nauplius stage.

### 2.6. EcVtg Genes Related to Ovarian Development of E. carinicauda

To confirm the role of *EcVtg* genes in the ovarian development of *E. carinicauda*, the expression profiles of *EcVtgs* at four typical stages were examined using qRT-PCR. As shown in Figure 6, all EcVtgs exhibited high expression levels in the hepatopancreas during the major-growth stage, which represents the primary growth stage. However, the expression patterns of EcVtgs differed across the other three stages of ovarian development. *EcVtg1a*, *EcVtg1b*, and *EcVtg3* displayed significantly higher expression in the mature stage compared to the minor-growth stage (*p* < 0.05). Conversely, *EcVtg8* exhibited significantly higher expression in the minor-growth stage than in the mature stage (*p* < 0.05). There were no significant differences between the minor-growth stage and mature stage in the expression of *EcVtg2*, *EcVtg4*, *EcVtg5*, *EcVtg6a*, *EcVtg6b*, and *EcVtg7*.

The mRNA expression of *EcVtgs* of the ovary during four ovarian-development stages are shown in Figure 7. The expression patterns of *EcVtg* genes were similar expect for *EcVtg5* and *EcVtg6a*, and their expression was highest in the major-growth stage, whereas *EcVtg6a* was highest in the mature stage (*p* < 0.05), and there was no difference in *EcVtg5* between the major-growth stage and the mature stage. Specifically, the mRNA expression of *EcVtg3* in the major-growth stage and the mature stage were thousands of times that of it in the proliferative stage and the minor-growth stage.

### 2.7. Expression of the EcVtgs after Eyestalk Ablation

Eyestalk ablation is an effective method of promoting ovarian development in crustaceans. The expression levels of *EcVtgs* in the hepatopancreas and ovaries after eyestalk removal were monitored using qPCR (Figure 8 and Figure 9). When compared with that of the control group with intact eyestalks, the expression levels of the *EcVtgs* in the hepatopancreas were significantly upregulated 6 and 11 days after eyestalk ablation. In contrast to those in the hepatopancreas, the expression levels of the *EcVtgs* in the ovaries were significantly upregulated 11 days after eyestalk ablation, and only the expression levels of *EcVtg1a*, *EcVtg2*, and *EcVtg3* in the ovaries in the day 11 group were higher than those in the control group, whereas the other *EcVtgs* exhibited no significant change in expression at 6 days after eyestalk ablation.

## 3. Discussion

Vtg is a precursor of yolk proteins in egg-laying vertebrates and invertebrates and has been found to have multiple forms in many fish species, which play important roles in vitellogenesis and embryonic development [24]. Zebrafish have at least seven distinct and functional *Vtg* genes [25], whilst three *Vtg* genes have been found in *Larimichthys crocea* [24], and two *Vtg* genes have been identified in *Opsariichthys bidens*, suggesting that different species have a variety of *Vtg* genes. Similar to various oviparous vertebrates, multiple *Vtg* genes are present in single-decapod crustacean species. As Dendrobranchiata species, *Fenneropenaeus merguiensis* [26] and *Litopenaeus vannamei* [27] have three *Vtg* genes and *Penaeus japonicus* has two *Vtg* genes [28]. Furthermore, as Pleocyamata species, four *Vtg* genes were identified in the *Procambarus clarkii* species [29], and seven *Vtg* genes were identified from *M. rosenbergii* [22] and *Macrobrachium nipponense*. We identified 10 different *Vtgs* in *E. carinicauda* using homology comparison, which were much more abundant than those in other crustaceans. These results also suggest that the number of *Vtgs* in Pleocyemata species is higher than that in Dendrobranchiata species. Liang et al. found a *Vtg* (AFM82474.1) in *E. carinicauda* which shares 99.10% identity tandemly with *EcVtg1a*. In this study, we comprehensively analyzed the 10 *EcVtg* genes, including their chromosome location, collinear relationships, gene structure, conserved motifs, and expression patterns.

In most oviparous animals, *Vtg* typically comprises three conserved domains: the N-terminal domain, a domain of unknown function (1943), and the von Willebrand factor type D domain [30]. SMART analysis identified these three conserved domains in all *EcVtgs*. Initially recognized for its involvement in vitellogenesis and embryo development, *Vtg* is increasingly acknowledged as a multifunctional protein contributing to non-reproductive functions, including innate immunity and antioxidation [31,32]. The N-terminal domain represents a conserved region acknowledged as the primary phosphorylation site and protein-modification region, crucial for *Vtg* cleavage, recognition by *Vtg* receptors, and the transportation of lipids and proteins [33]. *Vtg* is an antibacterial agent with a wide spectrum of functions in insects [34], which also increase the lifespan of honeybees by enhancing their oxidative-stress tolerance and shielding them from oxidative damage [35]. Additionally, *Vtg* plays a critical role in protecting *Eriocheir sinensis* from bacterial infections by inhibiting bacterial proliferation and regulating the expression of antimicrobial peptides [36]. As the DUF1943 and VWD domains in *Vtg* always interact with and remove pathogenic microorganisms, such as viruses or bacteria, it is proposed that *Vtg* is primarily a pattern recognition molecule and plays an important role in immunity. This indicated that the possibility of multifunctional function of *EcVtgs* occur in *E. carinicauda*.

The *EcVtg* genes identified in the *E. carinicauda* genome displayed minor variation in their sequence structures. Regarding protein length, the range of variation observed in the amino acid sequence of *EcVtgs* was from 2421 aa to 2581 aa. Concerning gene structure, the variation range of the *Vtg* gene exon spanned from 15 to 17. Furthermore, variations within the conserved motifs of *Vtg* genes ranged from 26 to 30. Typically, a 15-exon *Vtg* gene has been reported in shrimp species, including *M. ensis* [37], *P. monodon* [38], *Homarus americanus* [39], and *P. japonica* [15]. Across these reports, the sizes and positions of exons and introns within the *Vtg* gene remained consistent, indicating a high level of conservation among decapod *Vtg* genes. However, in the case of *E. carinicauda*, *EcVtg3* and *EcVtg7* contained 16 exons, and *EcVtg5* contained 17 exons. Similarly, *Vtgs* observed in *M. rosenbergii* also exhibit structures containing either 15 or 16 exons. The inquiry of whether similar *Vtg* gene organizations occur in other decapods necessitates further investigation. The different gene structures of *EcVtgs* suggest that their functions are diverse.

*EcVtg1a* and *EcVtg1b* share 91.51% of identity tandemly located on chromosome 23, and *EcVtg6a* and *EcVtg6b* share 97.92% of identity tandemly located on chromosome 10, while *EcVtg3–6* were present in a tandem cluster on chromosome 10. Gene-duplication analysis revealed that the *EcVtgs* family have expanded with duplication events in evolution. The Ka/Ks ratio of all repetitive *EcVtg* gene pairs were significantly less than 1.0 (ranged from 0.1981 to 0.4303). The ratio is of the number of nonsynonymous substitutions per nonsynonymous site (Ka) to the number of synonymous substitutions per synonymous site (Ks). The Ka is much less than Ks (i.e., Ka/Ks << 1) which means that selection keeps the protein as it is (purifying selection) [40]. This suggests that repetitive *EcVtg* genes in *E. carinicauda* were predominantly subjected to purifying selection during evolution.

*Vtg* is synthesized and secreted from the liver of oviparous vertebrates [41] and from fat bodies in insects [42], and is then taken up for oocyte development and other functions. In crustaceans, *Vtg* is produced mainly in the hepatopancreas and ovaries, deposited in the hemolymph, and then accumulated in growing oocytes. Various *Vtg* synthesis sites have been identified in different crustacean species. Several studies have suggested that the hepatopancreas is the principal site of *Vtg* synthesis in Pleocyemata [43,44], and only a small amount of *Vtg* is synthesized in the ovaries. In Dendrobranchiata, *Vtg* synthesis occurs in both the hepatopancreas and ovaries [45]. As shown in Table 3, the hepatopancreas and ovaries were found to be the principal site of *Vtg* synthesis in all Penaeidae shrimp species, including *L. vannamei*, *P. japonicus*, and *Penaeus monodon* [28,46,47]; the hepatopancreas was the principal site of *Vtg* synthesis in most Pleocyemata species. However, other studies have demonstrated that different *Vtg* isoforms are differentially expressed in the hepatopancreas and ovary, contradicting the past understanding of the *Vtg* synthesis site. *Vtg2* is primarily expressed in the hepatopancreas, whereas *Vtg1* and *Vtg3* are primarily expressed in the ovaries of *F. merguiensis* [26]. *Vtg1* is expressed primarily in the hepatopancreas and ovaries, but only *Vtg2* is expressed in the hepatopancreas of *M. ensis* [14]. Recently, some studies have reported the dual expression (in the hepatopancreas and ovaries) of *Vtgs* in species such as *M. nipponensis* [48], *P. clarkii* [29], and *Scylla paramamosain* [49].

In the present study, all *EcVtgs* were highly expressed in the hepatopancreas, whereas *EcVtg3*, *EcVtg4*, and *EcVtg6a* were simultaneously expressed in both the ovaries and hepatopancreas. This indicates that, whilst the ovaries are a synthesis site of *Vtg*, the hepatopancreas is the main site of *Vtg* synthesis in *E. carinicauda*. Liang et al. also found that *Vtg* (AFM82474.1), which includes *Vtg1a*, is mainly expressed in the hepatopancreas [19]. Therefore, exogenous *Vtg* is the major source of Vtg, which is synthesized in the hepatopancreas, transported via the hemolymph, and absorbed by oocytes or follicle cells via receptor-mediated endocytosis. The expression pattern indicated that all *EcVtgs* play an important role during the ovarian development of *E. carinicauda*; meanwhile, *EcVtg1a*, *EcVtg2* and *EcVtg3* play major roles in exogenous vitellogenesis, and *EcVtg3* plays a major role in endogenous vitellogenesis.

In most shrimp species, the synthesis of *Vtg* in the ovaries and hepatopancreas during ovarian development is regulated by the X-organ/sinus gland complex situated in the paired eyestalks. Unilateral eyestalk ablation in female shrimp is commonly practiced to induce ovarian maturation [64,65]. Removal of the eyestalk, observed in both *Pleocyamata* and *Dendrobranchiata*, has been shown to accelerate ovarian development and elevate the gonadosomatic index [66,67,68]. In the *Dendrobranchiata* shrimp species *P. japonicus*, mRNA expression of *Vtg1* notably increased in the ovaries while remaining unchanged in the hepatopancreas following eyestalk ablation, whereas *Vtg2* expression differed [28]. Bilateral ablation of the eyestalk significantly upregulated the mRNA expression of all *Vtgs* in the female hepatopancreas of *M. rosenbergii* but did not affect the expression of *Vtgs*, except *Vtg1* in the ovaries [22]. Eyestalk ablation is commonly used to promote ovarian development and maturation in *E. carinicauda* [69], which can promote synchronous ovarian development and shorten the breeding cycle of seedlings. We previously found that the gonadosomatic index increased after eyestalk ablation. On the 6th and 11th days after eyestalk ablation, the gonadosomatic index increased approximately three- and nine-fold when compared with the control group, respectively [70]. In the current study, the expression levels of all *EcVtgs* in the hepatopancreas exhibited significant upregulation at 6 and 11 days following eyestalk ablation. However, the expression levels of most EcVtgs in the ovaries were significantly upregulated specifically at the 11-day mark post eyestalk ablation. These indicated that removal of the eyestalk firstly promotes the ovarian development of *E. carinicauda* from exogenous, and next from endogenous.

## 4. Materials and Methods

### 4.1. Genome-Wide Identification and Sequence Analysis of Vtg Genes in E. carinicauda

The genome sequence, gene annotation, and gene and protein sequences were obtained from our previous study (unpublished, the BioProject was PRJNA1055619). To identify the candidate *Vtg* genes, we used a HMM to search the protein database for *E. carinicauda* genome using the hmmsearch software in HMMER v3.0, and, in which, the HMM profiles of the conservative functional domain of the lipoprotein amino-terminal region (PF01347), vitellinogen, open beta-sheet (PF09172), and von Willebrand factor type D domain (PF00094) were used as queries with default parameters. Then, the CD-Search Tool (https://www.ncbi.nlm.nih.gov/Struture/bwrpsb/bwrpsb.cgi, National Library of Medicine, Bethesda, MD, USA, accessed on 20 May 2022) was used for further verification following manual screening of redundant genes. The basic physicochemical properties of the *Vtg* proteins, including the amino acid number, MW, and pI, and they were predicted using the ExPASy tool (http://www.expasy.ch/tools/pi_tool.html, Swiss Institute of Bioinformatics, Lausanne, Switzerland, accessed on 5 June 2022).

### 4.2. Phylogenetic Analysis and Molecular Evolution

The phylogenetic trees of *Vtg* genes were constructed using MEGA11 software. These *Vtg* protein sequences were then utilized to generate a maximum likelihood phylogenetic tree through MEGA11. The best model of the phylogenetic tree was calculated and constructed, employing the pairwise-deletion option with MEGA11, and then a phylogenetic tree of all *Vtg* proteins was constructed using the maximum-likelihood method with 1000 bootstrap replicates. Finally, the phylogenetic tree was visualized and beautified using the online tool iTOL (https://itol.embl.de/, Biobyte Solutions GmbH, Heidelberg, Germany, accessed on 20 August 2022).

KaKs_Calculator 2.0 (https://sourceforge.net/projects/kakscalculator2/, SourceForge, San Diego, CA, USA, accessed on 10 June 2022) [71] was employed to calculate the non-synonymous (Ka) and synonymous (Ks) substitution rates, along with the Ka/Ks ratio, for each duplicated *Vtg* gene.

### 4.3. Chromosome Distribution, Gene Structure, and Protein Motif Analyses of EcVtg Genes

The Multiple Expectation Maximization for Motif Elicitation (MEME) online program (https://meme-suite.org/meme/index.html, University of Nevada, Reno, NV, USA, accessed on 15 June 2022) (5.5.3) was performed to identify conserved motifs of *EcVtg* proteins, with the number of motifs set to 30, the maximum length set to 100, and the other parameters set to default.

To elucidate the exon–intron structure of *Vtg* genes within the *E. carinicauda* genome, data regarding exon and intron positions were extracted from the genome Generic Feature Format (GFF) file. Finally, the chromosome distribution, gene motif, and exon–intron structure of *Vtgs* were visualized using TBtools.

### 4.4. Subcellular-Localization and Protein-Structure Prediction

The subcellular localization of *Vtg* in *E. carinicauda* was predicted using the online Wolf PSORT tool (https://www.genscript.com/wolf-psort.html, GenScript Biotech Corporation, Nanjing, China, accessed on 23 June 2022). Additionally, the secondary structure of the *Vtg* proteins was predicted using online SOPMA software (https://npsa-prabi.ibcp.fr/cgi-bin/npsa_automat.pl?page=npsa_sopma.html, Institute for the Biology and Chemistry of Proteins, Lyon, France, accessed on 20 June 2022) with the number of conformational states set to four, the similarity threshold set to eight, and the window width set to 17. Signal peptides were predicted using online SignalP 5.0 software (https://services.healthtech.dtu.dk/services/SignalP-5.0/, Department of Health Technology, Lyngby, Denmark, accessed on 27 June 2022).

### 4.5. Expression Analyses of EcVtg Genes

To illustrate the expression patterns of *EcVtg* genes in various tissues and during different ovarian-development stages, healthy female adults of *E. carinicauda* (with a body length of 51.72 ± 1.47 mm and body weight of 1.78 ± 0.21 g) were obtained from the Yellow Sea near Rizhao City, Shandong Province, China. For tissue-specific expression determination, RNA isolation involved collecting muscle, stomach, eyestalk, heart, intestine, ovary, gill, and hepatopancreas tissues. From eighteen shrimps, three samples from each tissue type were collected (six shrimp pooled together).

According to the criteria outlined in Wang’s study for ovarian-development stages, females were evaluated daily by observing the size and color of their gonads. Subsequently, four females in distinct ovarian developmental stages were selected. A total of seventy-two shrimp (six individuals × four stages × three replicates) were collected to analyze the expression patterns of *Vtg* across four ovarian developmental stages.

For the eyestalk ablation experiments, 36 females underwent bilateral eyestalk ablation, while 12 females were designated as the control group. Hepatopancreatic and ovarian tissues were collected at 1-, 6-, and 11-days post eyestalk ablation, with three parallel samples obtained for both the experimental and control groups, respectively.

### 4.6. Quantitative RT-PCR Analysis

Total RNA was extracted using a TransZol Up Plus RNA Kit (Trans, Beijing, China) according to the manufacturer’s standard protocol. mRNA was reverse transcribed using HiScript III RT SuperMix for qPCR expression analysis. To investigate mRNA expression of *EcVtgs*, the qPCR assay was performed using the ChamQ SYBR Color qPCR Master Mix (Vazyme, Nanjing, China) with an ABI PRISM 7500 Sequence Detection System (Applied Biosystems, Carlsbad, CA, USA). The following conditions were used for PCR: one cycle at 95 °C for 30 s, then 40 cycles of 95 °C for 10 s, 60 °C for 30 s, followed by one cycle of 95 °C and 15 s, 60 °C for 1 min, and 95 °C for 15 s.

β-actin derived from *E. carinicauda* (GenBank accession number: JQ045354.1) served as the internal control for the expression analysis of *E. carinicauda*. This particular gene was selected due to its abundance and stability within cells, exhibiting minimal susceptibility to external regulatory influences [72]. The primers for RT-PCR are shown in Appendix A. Real-time PCR assays were performed in triplicate, and the analysis was based on the CT values of the PCR products. The expression levels of *EcVtg* were calculated utilizing the 2^−ΔΔCT^ method. Statistical significance regarding differences in gene expression was determined using one-way ANOVA alongside Tukey’s multiple comparison tests, with a significance threshold set at *p* values < 0.05.

### 4.7. Expression-Pattern Analysis of EcVtg Genes during Embryonic Development

To investigate the expression patterns of *EcVtg* genes across six critical embryonic stages, encompassing cleavage, blastula, gastrula, nauplius, protozoa, and metazoea, the initial stage of the metazoea was collected following the methodology outlined in Wang’s study [73]. Total RNA extraction was conducted using the previously described method. Subsequently, a total of 21 libraries were generated, comprising three replicates from each of the seven groups. These libraries were sequenced using an Illumina HiSeq 4000 (San Diego, CA, USA) platform, generating 150 bp paired-end raw reads. The clean reads obtained from each RNA-seq library were aligned to the *E. carinicauda* genome utilizing STAR [74]. Differential expression analysis between the two groups was executed using the DEseq R package, employing criteria involving |log2fold change (FC)| ≥ 1 and a false discovery rate of <0.01. Heat maps illustrating the expression levels of the *EcVtg* genes were generated and visualized using the Heatmap Illustrator program within the TBtools.

## 5. Conclusions

The *EcVtg*-family genes were meticulously and systematically characterized in this study. A total of 10 *EcVtg* genes were identified within the genome of *E. carinicauda*, marking the most extensive number of *Vtgs* found in crustaceans so far. Phylogenetic analysis revealed that the *Vtg* phylogenetic tree aligned with the traditional taxonomic structure, illustrating the subdivision of the crustacean group into four distinct clusters: Astacidea, Brachyra, Penaeidae, and Palaemonidae. Molecular evolutionary analysis indicated that the evolution of *EcVtg* genes primarily underwent purifying selection. Furthermore, the qRT-PCR results demonstrated notably higher expression of all *EcVtgs* in the female hepatopancreas compared to other tissues. *EcVtg1a*, *EcVtg2*, and *EcVtg3* play major roles in exogenous vitellogenesis, and *EcVtg3* also plays a major role in endogenous vitellogenesis. The stimulation of ovarian development following eyestalk ablation was evidenced by the induction of *EcVtgs* expression in the hepatopancreas and, subsequently, in the ovaries, albeit at a later stage. This indicates that the hepatopancreas serves as the primary site for *Vtg* synthesis. The observed differential expression patterns among the *EcVtg* genes suggest a diverse array of functions during ovarian development in *E. carinicauda*. The differential expression patterns of various *EcVtg* genes in different tissues and during ovarian development indicates that their functions are diversified in *E. carinicauda*.

## Figures and Tables

**Figure 1 ijms-25-01089-f001:**
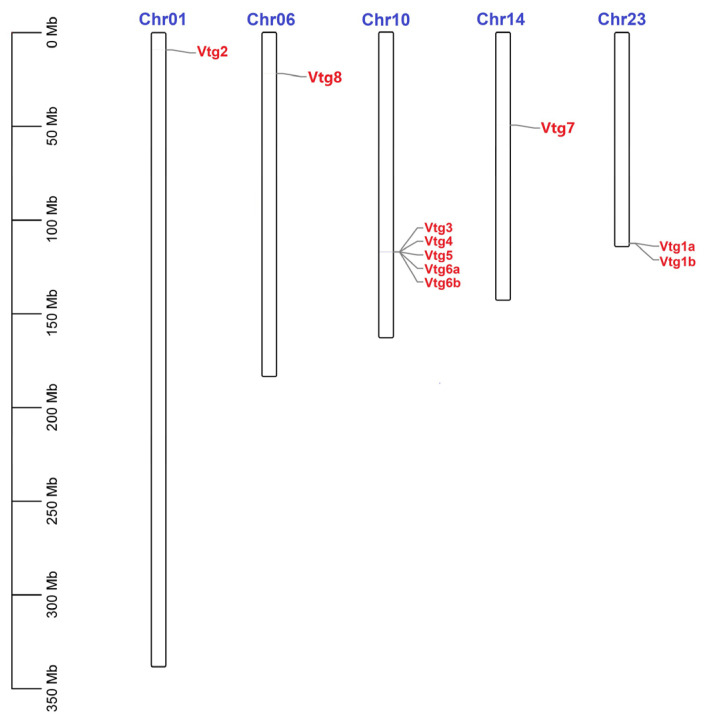
Chromosomal distribution of *EcVtgs* in *E. carinicauda*. The scale on the left was used to indicate the length of the chromosome. The bars refer to a total of five chromosomes. The corresponding gene names are written on the right side of the chromosomes.

**Figure 2 ijms-25-01089-f002:**
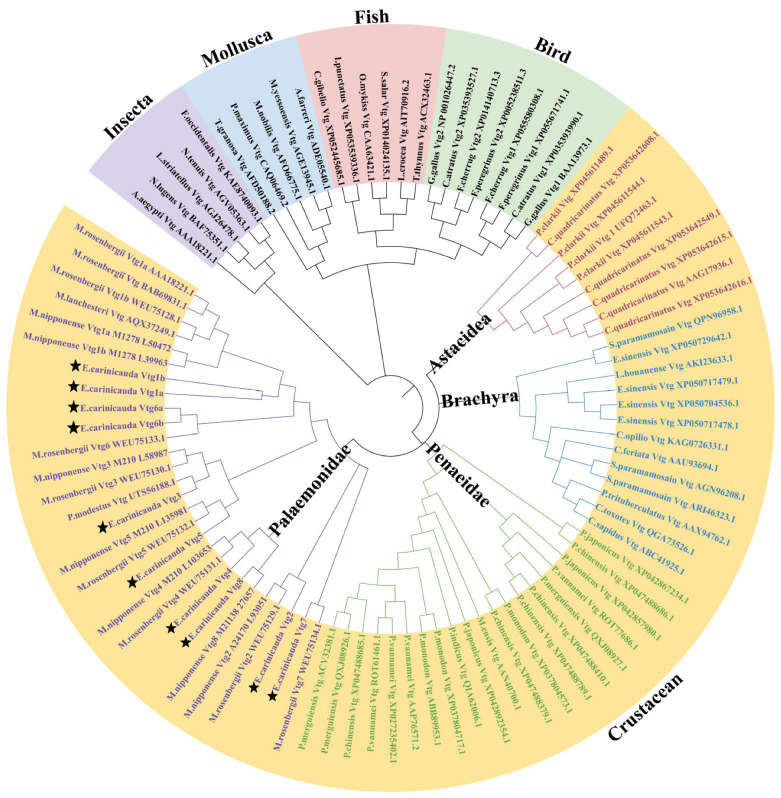
Phylogenetic analysis of Vtgs in representative species. The different colors represent five groups: purple, insecta; blue, mollusca; pink, fish; green, bird; yellow, crustacean. Different font colors represent four subgroups in crustacean; purple, Palaemonidae; green, Penaeidae; blue, Brachyra; red, Astacidea. The Vtgs of *E. carinicauda* were marked by star. Gene IDs and the full sequence of 94 *Vtg* proteins are listed in Appendix A.

**Figure 3 ijms-25-01089-f003:**
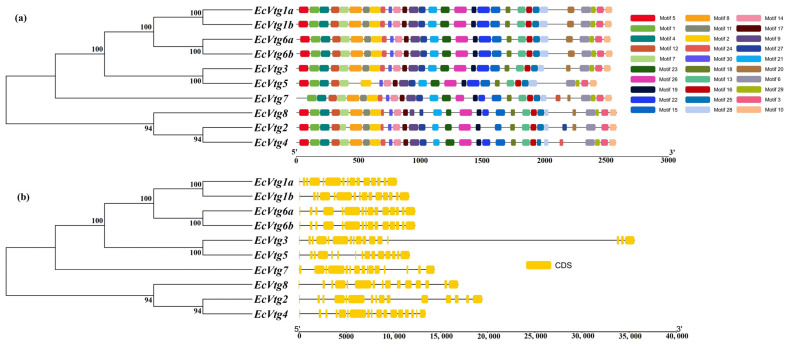
A schematic diagram of gene structure and conserved motif of the *EcVtg* gene family. (**a**) The conserved motifs of *EcVtgs*. The thirty conserved motifs are displayed in different colors and correspond one to one in the structural diagram. (**b**) The gene structures of *EcVtgs*. The yellow box represents the coding sequence; the black line represents the intron.

**Figure 4 ijms-25-01089-f004:**
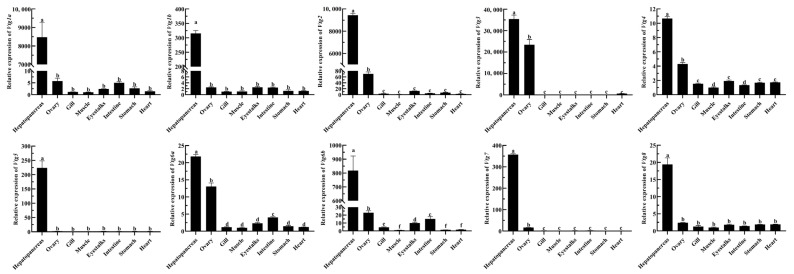
Relative abundance of *EcVtg* mRNA transcript in eight tissues. All experiments were performed independently at least three times. Error bars represent the standard deviation of three replicates. Different lowercase letters indicated significant differences in the different tissues (*p* < 0.05). The β-actin serves as an internal reference gene.

**Figure 5 ijms-25-01089-f005:**
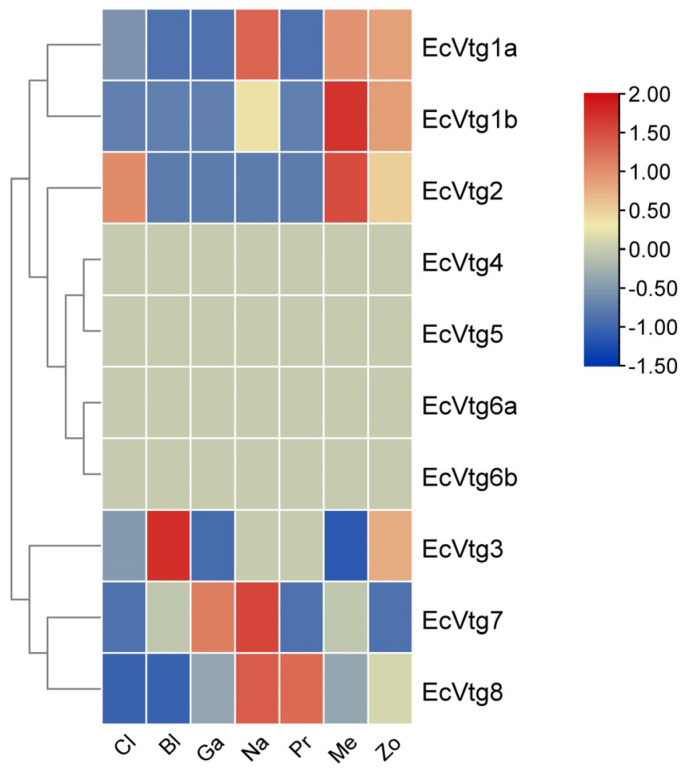
Heat map of differential expression of *EcVtg* genes during embryonic development. Cl, cleavage; Bl, blastula; Ga, gastula; Na, nauplius; Pr, protozoea; Me, metazoea; Zo, the first stage of zoea.

**Figure 6 ijms-25-01089-f006:**
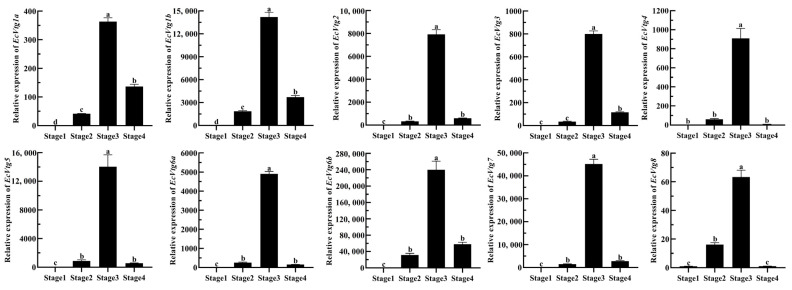
Relative abundances of *EcVtg* mRNA transcripts in hepatopancreas during the ovarian development. Stage 1, as the proliferative stage, the ovary is small and completely transparent when anatomically observed, and its morphology and color cannot be distinguished with in vitro observation. Stage 2, as the minor-growth stage, the ovary is enlarged, translucent with small black spots over its outer membrane, and positioned above the heart. Stage 3, as the major-growth stage, ovarian volume continues to increase, and its length has reached the 1/2 cephalothorax length, and the ovary is faintly yellow with small black dots over its ovarian membrane. Stage 4, as the mature stage, the ovary covers almost the entire stomach, hepatopancreas, and heart. All experiments were performed independently at least three times. Error bars represent the standard deviation of three replicates. Different lowercase letters indicated significant differences in the different stages (*p* < 0.05). The β-actin serves as the internal reference gene.

**Figure 7 ijms-25-01089-f007:**
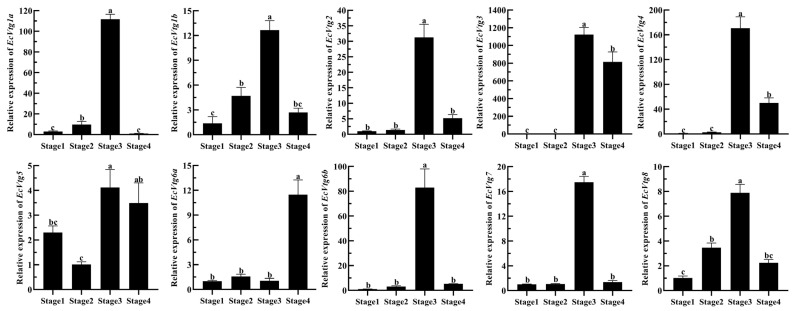
Relative abundances of *EcVtg* mRNA transcripts in the ovary during ovarian development. All experiments were performed independently at least three times. Error bars represent the standard deviation of three replicates. Different lowercase letters indicated significant differences in the different stages (*p* < 0.05). The β-actin serves as the internal reference gene.

**Figure 8 ijms-25-01089-f008:**
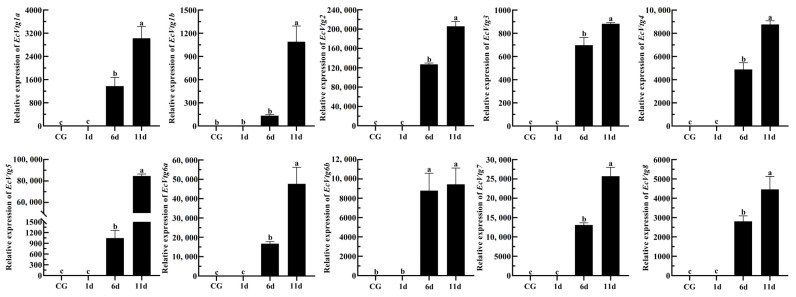
Expression level of *EcVtgs* after eyestalk ablation in the hepatopancreas. CG represents the control group with an intact eyestalk; 1d, 6d, and 11d represent the days after eyestalk ablation. All experiments were performed independently at least three times. Error bars represent the standard deviation of three replicates. Different lowercase letters indicate significant differences in the different stages (*p* < 0.05). The β-actin serves as the internal reference gene.

**Figure 9 ijms-25-01089-f009:**
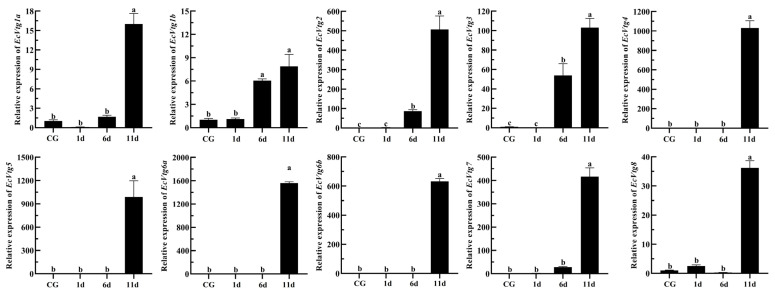
Expression level of *EcVtgs* after eyestalk ablation in ovary. CG represent the control group the control group with intact eyestalk, 1d, 6d and 11d represent the days after eyestalk ablation. All experiments were performed independently at least three times. Error bars represent the standard deviation of three replicates. Different lowercase letters indicate significant differences in the different stages (*p* < 0.05). The β-actin serves as the internal reference gene.

**Table 1 ijms-25-01089-t001:** Summary of sequence characteristics of *Vtg* genes in *E. carinicauda*.

Group	Gene	Chr	aa Number	MW (Da)	pI	Domain (From-To)
Vtg1	*Vtg1a*	23	2543	287.76	9.14	LPD_N (41-588), DUF1943 (623-917) VWD (2311-2451)
	*Vtg1b*	23	2543	287.30	9.02	LPD_N (43-588), DUF1943 (623-917) VWD (2311-2451)
Vtg2	*Vtg2*	1	2573	291.37	7.94	LPD_N (44-588), DUF1943 (621-910) VWD (2346-2488)
Vtg3	*Vtg3*	10	2533	284.05	9.03	LPD_N (42-586), DUF1943 (619-916) VWD (2298-2437)
Vtg4	*Vtg4*	10	2574	290.60	8.22	LPD_N (48-589), DUF1943 (622-925) VWD (2345-2484)
Vtg5	*Vtg5*	10	2421	275.06	7.74	LPD_N (45-521), DUF1943 (549-840) VWD (2191-2329)
Vtg6	*Vtg6a*	10	2530	283.70	7.72	LPD_N (51-598), DUF1943 (631-929) VWD (2298-2438)
	*Vtg6b*	10	2546	285.61	7.27	LPD_N (51-598), DUF1943 (631-929) VWD (2322-2454)
Vtg7	*Vtg7*	14	2573	289.13	7.86	LPD_N (77-597), DUF1943 (630-928) VWD (2335-2481)
Vtg8	*Vtg8*	6	2581	290.69	8.11	LPD_N (43-591), DUF1943 (624-922) VWD (2341-2487)

Chr, chromosome; aa, amino acid; MW, molecular weight; pI, isoelectric point.

**Table 2 ijms-25-01089-t002:** The predicted secondary structure and subcellular-location prediction of *EcVtg* proteins in *E. carinicauda*.

Gene	Alpha Helix	Extended Strand	Beta Turn	Random Coil	Subcellular Location
*EcVtg1a*	39.68%	19.90%	4.68%	35.75%	endoplasmic reticulum
*EcVtg1b*	39.44%	20.02%	4.44%	36.10%	extracellular matrix
*EcVtg2*	39.14%	19.78%	5.16%	35.92%	extracellular matrix
*EcVtg3*	39.76%	19.98%	4.90%	35.37%	extracellular matrix
*EcVtg4*	40.09%	19.54%	5.01%	35.35%	endoplasmic reticulum
*EcVtg5*	38.00%	20.16%	4.58%	37.26%	extracellular matrix
*EcVtg6a*	39.37%	19.84%	4.94%	35.85%	extracellular matrix
*EcVtg6b*	39.51%	19.84%	4.91%	35.74%	extracellular matrix
*EcVtg7*	38.63%	20.25%	5.25%	35.87%	endoplasmic reticulum
*EcVtg8*	39.05%	19.88%	4.69%	36.38%	nuclear

**Table 3 ijms-25-01089-t003:** The *Vtg* synthesis site in different crustaceans.

Species	Suborder	Family	Synthesis Site	Reference
*Exopalaemon carinicauda*	Pleocyemata	Palaemonidae	hepatopancreas	[19]
*Macrobrachium nipponensis*	Pleocyemata	Palaemonidae	hepatopancreas, ovary	[48]
*Macrobrachium rosenbergii*	Pleocyemata	Palaemonidae	hepatopancreas	[50]
*Pandalus hypsinotus*	Pleocyemata	Pandalidae	hepatopancreas	[51]
*Pandalopsis japonica*	Pleocyemata	Pandalidae	hepatopancreas	[52]
*Callinectes sapidus*	Pleocyemata	Portunidae	hepatopancreas, ovary	[53]
*Carcinus maenas*	Pleocyemata	Portunidae	hepatopancreas, ovary	[54]
*Charybdis feriatus*	Pleocyemata	Portunidae	hepatopancreas	[55]
*Portunus trituberculatus*	Pleocyemata	Portunidae	hepatopancreas	[51]
*Scylla paramamosain*	Pleocyemata	Portunidae	hepatopancreas, ovary	[49]
*Scylla serrata*	Pleocyemata	Portunidae	hepatopancreas	[56]
*Oziothelphusa senex senex*	Pleocyemata	Potamoidea	hepatopancreas	[57]
*Eriocheir sinensis*	Pleocyemata	Grapsidae	hepatopancreas, ovary	[43]
*Cherax quadricarinatus*	Pleocyemata	Parastacidae	hepatopancreas	[58]
*Procambarus clarkii*	Pleocyemata	Cambaridae	hepatopancreas, ovary	[29]
*Homarus americanus*	Pleocyemata	Palinuridae	hepatopancreas, ovary	[39]
*Upogebia major*	Pleocyemata	Upogebiidae	hepatopancreas, ovary	[44]
*Litopenaeus vannamei*	Dendrobranchiata	Penaeidae	hepatopancreas, ovary	[46]
*Metapenaeus ensis*	Dendrobranchiata	Penaeidae	hepatopancreas, ovary	[59]
*Penaeus chinensis*	Dendrobranchiata	Penaeidae	hepatopancreas, ovary	[60]
*Penaeus japonicus*	Dendrobranchiata	Penaeidae	hepatopancreas, ovary	[28,61]
*Penaeus monodon*	Dendrobranchiata	Penaeidae	hepatopancreas, ovary	[38,47]
*Penaeus semisulcatus*	Dendrobranchiata	Penaeidae	hepatopancreas, ovary	[62]
*Fenneropenaeus merguiensis*	Dendrobranchiata	Penaeidae	hepatopancreas, ovary	[26,63]

## Data Availability

Data are contained within the article and Appendix A.

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
