# Peer review of "Genome-Wide Identification of Vitellogenin Gene Family and Comparative Analysis of Their Involvement in Ovarian Maturation in Exopalaemon carinicauda"

_ijms, 2024, doi:10.3390/ijms25021089_

Round 1

Reviewer 1 Report

Comments and Suggestions for Authors

This study investigated the Vitellogenin (Vtg) genes in the ridgetail white shrimp,  Exopalaemon carinicauda,  a commercial mariculture species. Since Vtg genes play an important role in vitellogenesis and the reproductive success of oviparous species, the authors sought to identify and understand the function and tissue expression of multiple Vtg genes during the ovarian and embryonic development of the target species.
The article provides useful basic information for the species for the first time. However, to make it scientifically appealing to a broader audience, the following aspects need improvement or clarification:

1. The article's goals are not well-supported by adequate context (or hypotheses) in the introduction. For example, (lines 56-57) why does the analysis of these genes hold significant importance?

2. Likewise, in what context or working hypothesis did the study consider (74-75) the evolutionary relationships, gene structures, conserved domains, chromosomal locations, and motifs of the genes?

3. What was the underlying hypothesis (76-77) to investigate gene expression patterns in different tissues and at different ovarian stages?

4.  What would be the scientific or practical importance (considering the aquaculture value of the species) of some of these goals?

5. Section Results contains part of the discussion, i.e., other articles are cited/discussed.

6. The conclusion "Molecular evolutionary analysis indicated that the evolution of EcVtg genes primarily underwent purifying selection" is not discussed in the corresponding section (Discussion).

7. The conditional statement of the final conclusions should be replaced with more convincing arguments regarding the work done. For example,  "these genes potentially hold crucial roles in crustaceans". Or "this study provides relevant information for follow-up studies on the function of these genes". Follow-up studies were not clarified in the discussion.

8. Given the aquaculture importance of the species, the practical contribution of eyestalk removal should be better explained in the discussion.

Reviewer 2 Report

Comments and Suggestions for Authors

The authors have investigated and characterized 10 Vitellogenin (ECVtg) genes in Exopalaemon carinicauda. EcVtg genes were unevenly distributed on the chromosomes of E. carinicauda. In all putative EcVtg proteins, three conserved functional domains were found, including a lipoprotein N-terminal domain (LPD_N), a domain of unknown function (DUF1943), and a von Willebrand factor type D domain (vWD). Higher expression was seen in the female hepatopancreas compared to other tissues. EcVtg1a, EcVtg2 and EcVtg3 were proposed to play major roles in exogenous vitellogenesis, and in addition, EcVtg3 in endogenous vitellogenesis. The X-organ/sinus gland complex may play an important role in ovarian development, mostly by inducing Vtg synthesis.

Comments:

The genome sequence, gene annotation, gene and protein sequences were obtained from our previous study (unpublished): these data are not provided with this manuscript.

So, it hard to prove and compare the sequence data with other species. Are these data available in the database.

Can you give the sources of the the reference genome for Exopalaemon carinicauda and also give some information to its reference genome.

It is hard to differentiate what are new findings and which may be already known. The authors have to make clear their new findings, because they use data which are unpublished and online repositories.

In the introduction the state of the reference geome and proteom should be outlined. Which databases include this species and also its related species.

In the results section should be clearly indicated which results were obtained using bioinformatic tools and which one by own lab experiments.

Table S3: there a few typos. Could you also give the genus, family and order for the respective species.

KaKs_Calculator 2.0: please provide the source and version.

Comments on the Quality of English Language

No comments

Round 2

Reviewer 1 Report

Comments and Suggestions for Authors

Criticism and suggestions were carefully corrected, so the article is much improved and acceptable for publication.